# Trends of Fixed-Dose Combination Antibiotic Consumption in Hospitals in China: Analysis of Data from the Center for Antibacterial Surveillance, 2013–2019

**DOI:** 10.3390/antibiotics11070957

**Published:** 2022-07-15

**Authors:** Haishaerjiang Wushouer, Lin Hu, Yue Zhou, Yaoyao Yang, Kexin Du, Yanping Deng, Qing Yan, Xiaoqiang Yang, Zhidong Chen, Bo Zheng, Xiaodong Guan, Luwen Shi

**Affiliations:** 1Department of Pharmacy Administration and Clinical Pharmacy, School of Pharmaceutical Sciences, Peking University, Beijing 100191, China; kaiser@pku.edu.cn (H.W.); michelle_hlin@pku.edu.cn (L.H.); zhouyuezhy@pku.edu.cn (Y.Z.); yyy211anne@163.com (Y.Y.); lavenderdkx@126.com (K.D.); 2International Research Center for Medicinal Administration, Peking University, Beijing 100191, China; 3Department of Clinical Pharmacology, National Institute on Drug Dependence, Peking University, Beijing 100191, China; deng311@bjmu.edu.cn; 4National Institute of Hospital Administration, National Health Commission of the People’s Republic of China, Beijing 100044, China; yaoshi66@126.com (Q.Y.); yaoshichina6@163.com (X.Y.); 5Department of Pharmacy, Shanghai Sixth People’s Hospital, Shanghai 200233, China; antibiotic@163.com; 6Institute of Clinical Pharmacology, Peking University, Beijing 100191, China; doctorzhengbo@163.com

**Keywords:** antibiotic consumption, fixed-dose combination, China, trend, surveillance data

## Abstract

**Background: Fixed-dose** combination (FDC) antibiotics can be clinically inappropriate and are concerning with regards to antimicrobial resistance, with little usage data available in low- and middle-income countries. **Methods:** Based on retrospective data from the Center for Antibacterial Surveillance, we investigated the consumption of FDC antibiotics in hospital inpatient settings in China from 1 January 2013 to 31 December 2019. The metric for assessing antibiotic consumption was the number of daily defined doses per 100 bed days (DDD/100BDs). FDC antibiotics were classified according to their composition and the Access, Watch, Reserve (AWaRe) classification of the World Health Organization. **Results:** A total of 24 FDC antibiotics were identified, the consumption of which increased sharply from 8.5 DDD/100BDs in 2013 to 10.2 DDD/100BDs in 2019 (*p* < 0.05) despite the reduction in the total antibiotic consumption in these hospitals. The increase was mainly driven by FDC antibiotics in the *Not Recommended* group of the AWaRe classification, whose consumption accounted for 63.0% (6.4 DDD/100BDs) of the overall FDC antibiotic consumption in 2019, while the consumption of FDC antibiotics in the *Access* group only accounted for 13.5% (1.4 DDD/100BDs). **Conclusion:** FDC antibiotic consumption significantly increased during the study period and accounted for a substantial proportion of all systemic antibiotic usage in hospitals in China. FDC antibiotics in the *Not Recommended* group were most frequently consumed, which raises concerns about the appropriateness of FDC antibiotic use.

## 1. Introduction

Rising antimicrobial resistance (AMR) is a global health crisis, with serious consequences for morbidity, mortality, and healthcare costs [1,2,3,4]. However, the development of new antibiotics, which can take a decade or longer, can hardly keep pace with the emergence and spread of AMR, making the economic use of currently available antibiotics necessary [5,6,7,8]. Fixed-dose combination (FDC) medications, which contain two or more active substances within a single dosage form, represent a strategy to circumvent much of the early phases of drug development and reduce the lengthy timeline [7,9]. FDCs can improve treatment response compared to monotherapy through synergistic mechanisms of action (such as sulfamethoxazole/trimethoprim) or by improving the medication adherence of patients [10]. FDCs are well-established in treating conditions such as tuberculosis, malaria, and HIV [11,12]. Recent research has indicated that the sensible use of pharmacodynamically synergistic antibiotics through FDC may delay or even prevent the emergence of AMR [13]. One of the prominent groups of these FDC antibiotics with broad-spectrum activity is the combination of β-lactam and β-lactamase inhibitor (BL-BLI), which is regarded as a potentially effective strategy for infections caused by drug-resistant Gram-negative bacilli [14,15].

However, FDC antibiotics can be clinically inappropriate, as reported in some countries [16,17,18]. Concerns have been raised due to a lack of efficacy, increasing toxicity, and their potential impact on resistance [19]. High consumption of FDC antibiotics was observed in some low- and middle-income countries (LMICs) where FDCs accounted for a substantial proportion of the total antibiotic consumption [11]. In the past two decades, China is among the countries consuming the most antibiotics in the world [20]. Hence, China has committed to confining AMR through a series of policies and made considerable progress in limiting its total antibiotic consumption [21,22]. Nonetheless, a recent study showed that the country remained the second largest consumer of FDC antibiotics among 75 countries in 2015 as measured by sales data [11], but the data on the trends and patterns of FDC consumption were limited. In this study, we aim to describe the trends and patterns of FDC antibiotic consumption in hospitals in China over seven years. The rest of this paper is organized as follows. The methodology of the study is presented in Section 2. Then, the detailed results of FDC antibiotic consumption are proposed in Section 3. In Section 4, we discuss the findings of the study. The conclusion of the study is presented in Section 5.

## 2. Materials and Methods

### 2.1. Study Design

This is an observational study investigating the consumption of FDC antibiotics in hospital inpatient settings in China from 1 January 2013 to 31 December 2019.

### 2.2. Data Source

Data were retrieved from the Center for Antibacterial Surveillance (CAS), the largest nationwide surveillance database collecting data on antibacterial use from 31 (out of 34) provinces in China (excluding Hong Kong, Macao, and Taiwan). All secondary and tertiary hospitals that reported data on antibacterial use to the CAS database during the study period were included in the analyses. The specific numbers of hospitals included each year varied between 1630 in 2013 and 2486 in 2019. Hospital characteristics are shown in Table A1. The database collected quarterly data on inpatient antibacterial use. The CAS Quality Control Committee audited the data quality to ensure the data met the relevant requirements.

### 2.3. Data Collection

We extracted data on the annualized consumption of antibiotics and aggregated data at the level of active substances. The CAS database contained information about the following variables: region, hospital name, year, drug specification, unit, active substance designation, the quantity of consumption, and bed days.

We defined FDC antibiotics as medications consisting of two active substances, in which at least one was an antibiotic under the Anatomical Therapeutic Chemical (ATC) classification J01, as recommended by the World Health Organization (WHO) Collaborating Centre for Drug Statistic Methodology [23]. FDC antibiotics were then classified into antibiotic plus antibiotic adjuvant (A and A) or dual antibiotics (DA) according to the composition for further analysis [7,11,16].

### 2.4. Measurement

#### 2.4.1. Indicators

Antibiotic consumption was expressed as the number of defined daily doses per 100 bed days (DDD/100BDs) based on the WHO Collaborating Center for Drug Statistic Methodology [24,25]. For drugs that could not be coded according to the ATC system, the dosage regimen recommended in the manufacturers’ instructions was used as an alternative to gauge their DDD, as approved by the Chinese Food and Drug Administration [26].

We calculated the compound annual growth rate (CAGR) [27] of the consumption of FDC antibiotics to calculate a comparable metric across time.
(1)$$\mathrm{CAGR}\;={\left(\frac{C_{2019}}{C_{2013}}\;\right)}^{\frac{1}{6}}$$

*C*_2019_: Total FDC antibiotic consumption for the year 2019 (expressed as DDD/100BDs). 

*C*_2013_: Total FDC antibiotic consumption for the year 2013 (expressed as DDD/100BDs).

#### 2.4.2. WHO AWaRe Classification

FDC antibiotics were assessed based on the 2021 revision of the WHO Access, Watch, Reserve (AWaRe) classification [28]. The AWaRe classification was established based on the strength of the antibiotics and the potential impacts on antimicrobial resistance. Antibiotics in the *Access* group were the first- or second-line treatments for common infections and should be widely accessible. Antibiotics in the *Watch* group should only be used for a limited group of well-defined syndromes and under close surveillance. Antibiotics in the *Reserve* group should be primarily referred to as the last resort to treat infections caused by multi- or extensively drug-resistant bacteria. The fourth group, “*Not Recommended*”, consisted of antibiotic combinations whose use may negatively impact AMR and patient safety. In our study, we added a fifth group, “*Not Included*, to refer to antibiotics not included in the WHO AWaRe classification, but which were used in China.

### 2.5. Data Analysis

We assessed the consumption of FDC antibiotics in hospitals at a national level and comparatively analyzed the total antibiotic consumption to better understand the trend change. The consumption of FDC antibiotics at a provincial level was also analyzed to identify potential regional differences. Measures of relative consumption, expressed as a proportion of the total consumption of FDC antibiotics, were calculated based on FDC composition as well as the AWaRe classification for further analysis. Trends and ranking of the most frequently used FDC antibiotics were also assessed.

We computed linear regressions to assess the trends in the consumption of FDC antibiotics. The dependent variable was the consumption of FDC antibiotics, and the independent variable was time. Data were managed and analyzed in Microsoft Excel 2019 (Microsoft, Washington, DC, USA) and STATA 15.1 (StataCorp LLC, College Station, TX, USA). Figures were plotted using Origin (Pro), Version 2020b (OriginLab Corporation, Northampton, MA, USA). A difference of *p* < 0.05 was considered indicative of statistical significance.

## 3. Results

A total of 24 FDC antibiotic agents were identified in the CAS database from 1 January 2013 to 31 December 2019, including 7 DA agents and 17 A and A agents. Among these, four were not included in the AWaRe classification, fourteen contained agents categorized as *Not Recommended*, three contained agents in the *Access* group, two contained agents in the *Watch* group, and one contained agents in the *Reserve* group (Table A2).

### 3.1. Total Consumption of FDC Antibiotics 

From 2013 to 2019, the consumption of FDC antibiotics in hospital inpatient settings in China significantly increased from 8.5 DDD/100BDs recorded in 1630 hospitals in 2013 to 10.2 DDD/100BDs recorded in 2486 hospitals in 2019 (*p* < 0.05), with a CAGR of 3.0%. Meanwhile, the contribution of FDC antibiotics to the total consumption of antibiotics in these hospitals increased by 6.2% (*p* < 0.05). However, a contrasting trend was found in the total consumption of antibiotics in hospitals, which showed a significant decrease from 48.8 DDD/100BDs in 2013 to 43.0 DDD/100BDs in 2019 (Figure 1, Table A2).

### 3.2. Regional Distribution of FDC Antibiotic Consumption

Figure 2a presents the consumption of FDC antibiotics, stratified by region, in China in 2019. Xizang, Guangxi, and Jiangxi were the top three largest consumers of FDC antibiotics in 2019, with a consumption of 14.4 DDD/100BDs, 13.1 DDD/100BDs, and 12.8 DDD/100BDs, respectively. The consumption of FDC antibiotics substantially increased in most regions (28 provinces) across China, while only three provinces (Shanghai, Zhejiang, and Hunan) demonstrated a moderately decreasing trend between 2013 and 2019 (Figure 2b).

### 3.3. The Consumption of FDC Antibiotics in Different Compositions and AWaRe Classification

Of the two types of FDC antibiotics, A and A agents were more frequently consumed, contributing 10.0 DDD/100BDs (98.4%) to the total consumption of FDC antibiotics in 2019. Both DA and A and A agents demonstrated increasing trends in consumption over time (Figure 3a).

The proportion of the consumption of FDC antibiotics not included in the 2021 WHO AWaRe classification dropped from 0.3% to 0.1% during the study period. When assessing FDC antibiotics included in the AWaRe classification, agents in the *Not Recommended* group (6.4 DDD/100BDs) accounted for 63.0% of the consumption of all FDC antibiotics in 2019. The consumption of FDC agents in the *Access* group and the *Watch* group were 1.4 and 2.4 DDD/100BDs, accounting for 13.5% and 23.4%, respectively. There was a reduction in the proportion of the consumption of FDC antibiotics in the *Access* group (from 19.8% to 13.5%), while increasing trends were observed in the consumption of FDC antibiotics in the *Not Recommended* group (from 61.1% to 63.0%) and the *Watch* group (from 18.8% to 23.4%) (Figure 3b, Table A2).

### 3.4. The Consumption of Most Frequently Used FDC Antibiotics

Among the 24 FDC antibiotics identified, the consumption of only four FDCs (cefoperazone/sulbactam, piperacillin/tazobactam, mezlocillin/sulbactam, and cefotaxime/sulbactam) significantly increased from 5.2 DDD/100BDs in 2013 to 6.5 DDD/100BDs in 2019 and contributed to 63.4% of the total consumption of FDC antibiotics in hospital inpatient settings in China in 2019 (Table A2).

The seven most frequently consumed FDC antibiotics were cefoperazone/sulbactam, piperacillin/tazobactam, amoxicillin/clavulanic acid, mezlocillin/sulbactam, cefoperazone/tazobactam, piperacillin/sulbactam, and imipenem/cilastatin in both 2013 and 2019. These FDC antibiotics accounted for 90.6% of the total consumption of FDC antibiotics in China in 2019. Among these frequently consumed FDC antibiotics, significant increasing trends were observed in cefoperazone/sulbactam (*p* < 0.05), piperacillin/tazobactam (*p* < 0.05), and mezlocillin/sulbactam (*p* < 0.05), while amoxicillin/clavulanic acid was the only antibiotic that experienced a significant decrease during the study period (*p* < 0.05) (Table A2).

## 4. Discussion

To our knowledge, this study is the first to estimate the consumption of FDC antibiotics over time at a national level using surveillance data in China. In contrast to the reduction in the total consumption of antibiotics in hospital inpatient settings in China, the consumption of FDC antibiotics significantly increased between 1 January 2013 and 31 December 2019.

Overall, the consumption of FDC antibiotics accounted for a notable proportion (23.7% in 2019) of all systemic antibiotics in hospitals in China. Several studies estimated the FDC antibiotic consumption using sales data and showed varying findings on the FDC proportion. Our finding is lower than the proportion in India (33.8%) [16] but slightly higher than the proportion shown in a study conducted in eight Latin American countries (21.0%) [18] and a multinational study involving 75 countries (22.5%) [11]. Although these results cannot be directly compared due to the discrepancy between sales data and usage data, they, in some way, provided a reference for how frequently FDC antibiotics are used in China. The considerable proportion of FDC antibiotics shown in our study is mainly attributable to A and A agents, most of which were comprised of β-lactam and β-lactamase inhibitor (BL-BLI) (16/17). The potential reason for the extensive use of BL-BLI may be driven by increasing incidences of extended-spectrum β-lactamases (ESBLs) producing bacteria in hospital settings in China [29,30,31]. To confine the resistance of β-lactamase, medical institutions prefer the FDC of β-lactam antibiotics and β-lactamase inhibitors to improve their antimicrobial activity [32,33]. However, indiscriminate and widespread use of such combinations may significantly compromise patient care and trigger large-scale resistance development [34,35]. Therefore, further research is needed to investigate the safety and efficacy profiles of these BL-BLI combinations.

Previous studies raised concerns about the widespread use of unorthodox BL-BLI in India and about China flouting antimicrobial stewardship (ASP) and compromising patient care [34]. Unorthodox combinations may compromise clinical outcomes and potentially contribute to resistance development [35]. These FDC antibiotics have been introduced into clinical practice without mandatory drug development studies involving pharmacokinetic/pharmacodynamic, safety, and efficacy assessments being undertaken [36]. The Indian Government has already taken a series of initiatives to deal with this problem [37]. Despite the increasing concern of AMR, the usage of FDC antibiotics in China is still largely overlooked. Although in vitro studies were conducted [38,39,40], evidence on the safety and/or efficacy of FDC antibiotics is far from adequate [41,42] to support these unorthodox combinations that are categorized as *Not Recommended* by the AWaRe classification. Rather than imitating the composition of BL-BLIs through rigorous drug development, sulbactam or tazobactam has been arbitrarily combined with cephalosporins by indigenous drug manufacturers in China [34]. These unorthodox BL-BLIs may cause uncertain clinical outcomes or even clinical failure, resistance development, and drug toxicity [11,43]. Thus, more studies are required to evaluate their safety, efficacy, and effectiveness to promote the appropriate use of antibiotics.

In the past two decades, the Chinese Government has made various attempts to confine AMR through policies and measures, including limiting the usage of antibiotics [44]. However, the implementation of ASP overlooked the inappropriate use of FDC antibiotics [34]. A core element of the ASP, the Antibiotic Formulary Restriction (AFR), categorized antimicrobials into three classes (*Non-Restricted*, *Restricted*, and *Highly Restricted*) [21]. It was developed at a provincial level instead of a national level to accommodate for regional differences in the prevalence of bacteria resistance [45]. Our study finds that among the seven most frequently consumed FDC antibiotics that contributed 90% to the total consumption of FDC antibiotics in China, cefoperazone/sulbactam, mezlocillin/sulbactam, cefoperazone/tazobactam, and piperacillin/sulbactam were classified as *Restricted* in most provinces, though they were classified as *Not Recommended* according to the AWaRe classification. This discrepancy implies that these medications, despite having limited evidence on safety and efficacy profiles and thus, need to be used with caution, can be easily accessed in China. In addition, these FDC antibiotics were classified into levels of higher restriction in provinces such as Shanghai and Zhejiang, where the consumption of FDC antibiotics showed a negative CAGR during the study period. This indicates that in regions with a higher level of development, more medical resources, and a higher level of management, FDC antibiotics were used more prudently compared to less developed regions.

As globalization allows resistant micro-organisms to spread rapidly to distant countries and continents, the regulation of FDC antimicrobials in China has implications beyond the country itself [37]. A multisectoral approach involving all stakeholders is needed to curb the potentially inappropriate use of FDC antibiotics, and regulators should strengthen the enforcement of relevant regulations. The industry should also act responsibly, ensuring that the development of FDC antibiotics adheres to scientific standards and generates robust efficacy and safety data. Moreover, physicians should be better educated through formal education or training sessions about the public health implications of the inappropriate use of FDC antibiotics. Pharmacists can also help to address the inappropriate use of FDC antibiotics through their professional knowledge of drugs.

### Limitation

This study has several limitations. Firstly, the reporting of hospitals to the CAS database is on a voluntary basis, and each year the number of hospitals included in the CAS database varies, which might have introduced selection bias. However, since these hospitals included mostly tertiary hospitals and spanned most provinces in China, our study sample remained representative to a certain extent. Secondly, the CAS database mainly covered secondary and tertiary hospitals across China with a paucity of data from primary healthcare settings. As primary healthcare facilities generally had very limited inpatient capacity, our results could still reflect patterns in the consumption of FDC antibiotics in inpatient settings in China. Thirdly, only aggregated data, instead of hospital-level data, were available, and details regarding quality control were inaccessible. However, the CAS database conducts internal quality control measures. The National Health Commission also carried out training programs to improve data collection, which could ensure the quality of the data. Fourthly, due to limited data availability, we only analyzed the use of FDC antibiotics in inpatient settings. This might lead to an underestimation of the consumption of some antibiotics, such as amoxicillin–clavulanic acid and sulfamethoxazole–trimethoprim. However, as most FDC antibiotics are restricted in outpatient settings according to antibiotic formulary restriction management in China [46], our results therefore remain considerably representative of the trends and patterns of FDC antibiotics used in China.

## 5. Conclusions

Our study finds that the consumption of FDC antibiotics represented a substantial proportion of all systemic antibiotics in hospital inpatient settings in China, with a significant increasing trend despite the reduction in the total consumption of antibiotics. FDC antibiotics in the *Not Recommended* group were most frequently consumed, which might indicate potentially inappropriate use of these medications. More attention should be paid to establishing specialized management approaches promoting the appropriate use of FDC antibiotics to better confine AMR nationally and internationally.

## Figures and Tables

**Figure 1 antibiotics-11-00957-f001:**
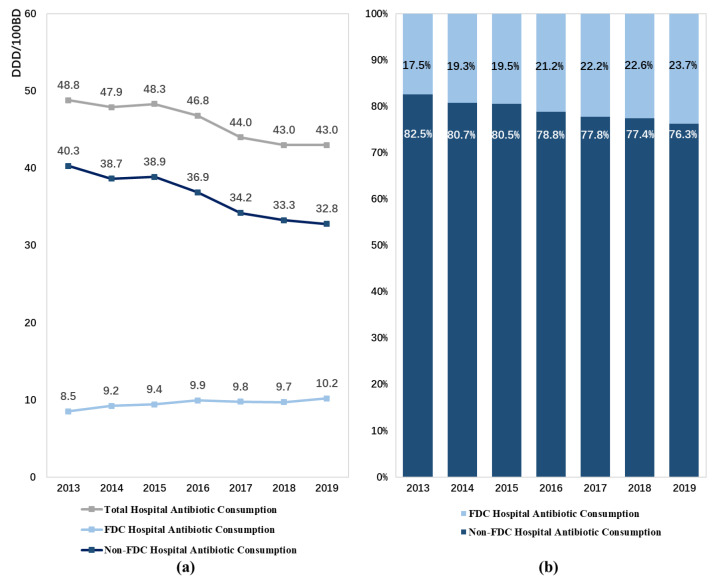
Patterns of fixed-dose combination antibiotic consumption in hospitals in China from 2013 to 2019: (**a**) expressed as the number of DDD/100 bed days, (**b**) expressed as percentage proportions. Abbreviation: FDC, fixed-dose combination.

**Figure 2 antibiotics-11-00957-f002:**
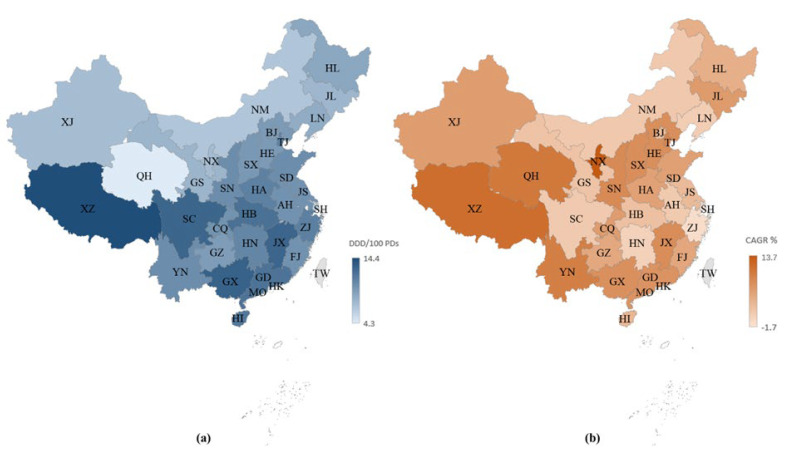
Fixed-dose combination antibiotic consumption in hospitals in China: (**a**) expressed as DDD/100 bed days in 2019, (**b**) expressed as compound annual growth rates between 2013 and 2019. Abbreviations: BJ, Beijing; SH, Shanghai; TJ, Tianjin; CQ, Chongqing; HE, Hebi; SX, Shanxi; NM, Inner Mongolia; LN, Liaoning; JL, Jilin; HL, Heilongjiang; JS, Jiangsu; ZJ, Zhejiang; AH, Anhui; FJ, Fujian; JX, Jiangxi; SD, Shandong; HA, Henan; HN, Hunan; GD, Guangdong; GX, Guangxi; HI, Hainan; SC, Sichuan; GZ, Guizhou; YN, Yunnan; XZ, Tibet; SN, Shaanxi; GS, Gansu; QH, Qinghai; NX, Ningxia; XJ, Xinjiang.

**Figure 3 antibiotics-11-00957-f003:**
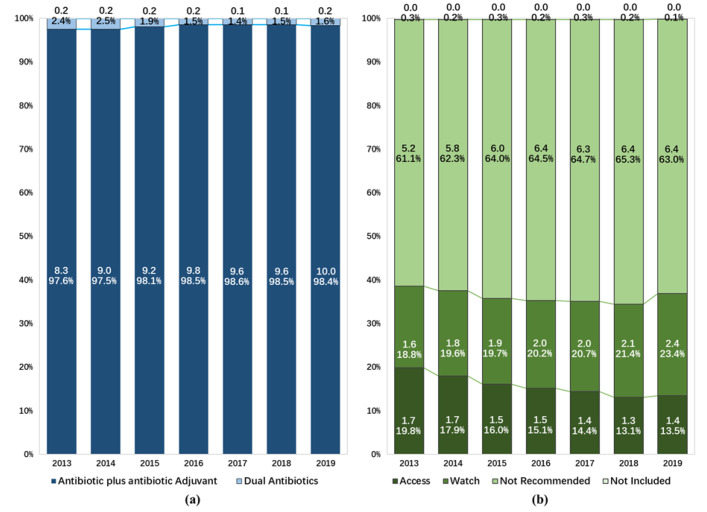
Composition of fixed-dose combination antibiotic consumption in hospitals in China from 2013 to 2019: (**a**) expressed as different types of chemical composition, (**b**) expressed as the Access, Watch, Reserve classification of the World Health Organization.

## Data Availability

The datasets used and analyzed during the current study are available from the corresponding author on reasonable request.

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
