# Peer review of "Trends of Fixed-Dose Combination Antibiotic Consumption in Hospitals in China: Analysis of Data from the Center for Antibacterial Surveillance, 2013–2019"

_antibiotics, 2022, doi:10.3390/antibiotics11070957_

Round 1
Reviewer 1 Report
This paper studies Trends of Fixed-Dose Combination Antibiotic use in China: data from the Center for Antibacterial Surveillance and 2013-2019. There are a few weaknesses that should be addressed in this paper. Therefore, I suggest the authors resubmit it after a major revision. My suggestions are as follows:
General comments:
1. The structure of your paper is so weird. Please consider the main structure of the paper at the end of the introduction.
2. Discuss more the limitations of the study and future research suggestions if there are any.
3. The paper should be revised to include at least 10 more recent references in 2021-2022.
4. Your section and subsections should be ordered in numbers based on the MDPI structure.
5. The quality of English needs to be improved across the paper. Also, the scientific terms pertinent to your topic should be improved.
Main comments:
1. In line 145 you have mentioned "Southern provinces used more FDC antibiotics than the northern region" Please explain more about it? what is the main reason behind this strategy? why do Southern provinces use more FDC antibiotics than the northern region? please explain more?
2. In line 153 you have mentioned "Among the two types of FDC antibiotics, A&A agents were more frequently used" Please explain more about it? what is the main reason behind this strategy? why A&A agents were more frequently used?
3. Please explain more about “Individual FDC antibiotic use” in line 166.
4. In line 184 you have mentioned "Overall, the use of FDC antibiotics accounted for a substantial proportion " this part is unclear. please explain more. please provide more reasons behind this conclusion.
Reviewer 2 Report
The manuscript deals with the "Trends of Fixed-Dose Combination Antibiotic use in China…." A compilation of data sets of fixed-dose combinations from 2013-to 2019 was conducted. The objective of the present study was to describe the trends and patterns of fixed-dose combination antibiotics used in China over seven years.
General comments: A very laborious manuscript. However, the background is a little descriptive. In addition, some statements should be better explained. The objective of the present study should be enriched. The data source should be thoroughly described in a way that leaves no room for doubt (see Specific Comments). If possible, replace the word agent with active substance and the word used by consumption.
This manuscript is not yet ready for publication.
Specific Comments
L1: The title should reflect the content of the manuscript. It is proposed the following title:
Trends of Fixed-Dose Combination of Antibiotic Consumption in Hospitals of China: Analysis of data from Center for Antibacterial Surveillance, 2013-2019.
L47-50: it should be improved. Increasing adherence to therapy (?)
L51-53: it should be improved. FDC antibiotics can be clinically inappropriate, as reported in some countries. (?)
L58-62: A recent study showed that China remained the second-largest consumer of FDC antibiotics among countries as measured by sales data. However, this manuscript aims to describe the trends and patterns of fixed-dose combination antibiotics used in China over seven years. What is the added value of this publication? How is it different from the published study?
L65: antibiotic use – I propose antibiotic consumption. Antibiotic hospital consumption in China (?)
L67-73: The data was collected considering the hospital's inpatients (I presume), and this data comprised 2161 secondary hospitals and 1980 tertiary hospitals (?). The data was retrieved from CAS; but quarterly data regarding inpatient antibiotic use was extracted (?).
L75: March…December… is just a quarterly event. I propose to erase quarterly data regarding inpatient antibiotic use was extracted.
L78: We extracted data on yearly antibacterial use (antibiotic consumption) aggregated at the level of the active substance. It is proposed to write antibiotic consumption
L80: generic name: active substance designation (?) - I think active substance designation will be better. Thus, there will be no confusion with the terms brand/generic medicinal products (in this case, antibiotics)
L81-82: The data on inpatient days were calculated by multiplying the quarterly total number of hospital discharges with the mean number of days of hospitalization. The objective and meaning of this statement (?) bed-days? Admissions?
L91-92: bed-days? The DDDs per 100 bed days are applied when drug use by inpatients is considered. This measure is applied in analyses of in-hospital drug use. Do the authors refer to PDs, as the mean number of days of hospitalization? Definition of patient days and references.
L102-104: In 2021 has been updated. Now the AWaRe classification includes an additional 78 antibiotics not previously classified, bringing a total of 258. https://www.who.int/publications/i/item/2021-aware-classification
L116: We assessed the FDC antibiotic hospital consumption (?) at the national level…
L131: 2021 (?)
L136: hospitals of China - inpatients? How many hospitals? Secondary hospitals? Tertiary hospitals?
L139: used in hospitals (?)
L145-149: The acronyms of the cities, as shown in Figure 2 should be placed in parentheses after the city name.
Figure 1: Total Antibiotic use = Total hospital antibiotic consumption; FDC hospital antibiotic consumption; ; Non-FDC hospital antibiotic consumption
…antibiotic consumption in inpatients hospitals of China (?)
L155: The % of DA should be included within the light blue colour. Dual antibiotics and not dual antimicrobials.
L166: instead of use, consumption
L185-187: The % values should be reviewed. Do they match similar permisses? Both consumption in hospitals? Both usage in inpatients? Will it be possible to compare them?
L191: Increasing Incidence of Extended-Spectrum β-Lactamase-Producing Bacteria in Community Hospitals (????)
L191: To overcome the resistance of ß-lactamase. It should be better written.
L195: EVIDENCE. Of what? More studies? Antibiotic effectiveness? This type of antimicrobial resistance has been recognized worldwide in the last 20 years. There is little data about the incidence of community-acquired infections caused by ESBL-producing bacteria. And in hospitals?
L197: Unorthodox combinations compromise clinical outcomes and potentially contribute to resistance development. Antibiotic combinations have been introduced into clinical practice without mandatory drug development studies involving pharmacokinetic/pharmacodynamic, safety, and efficacy assessments being undertaken. Pease all these possible constraints should be discussed considering the unorthodox combinations …
L209: … their safety, efficacy, effectiveness (considering the antibiotic medicine development)
L249: sulfamethoxazole methotrexate - Methotrexate is not an antibiotic
L255: instead of use, consumption
Reviewer 3 Report
This study aims to describe the trends and patterns of 61 FDC antibiotics used in China over a 7-year period. This manuscript does not fulfill the standards established for the journal to be considered for publication.
- The manuscript needs to be revised for English.
What are DDD/100PDs? add the full name in the abstract.
- Introduction is very brief and not informative. For example, you need to talk about the most common FCD in china and what infections they used to treat.
- In the data source section, you need to add a table to distribute the collected data of 4141 over the sampling area.
- Where is the statistical analysis to show the significance of the results.
- Further analysis is required as the authors talk all the time about FCD ut they did not mention these antibiotics combinations, what pathogens are treated with different combination
- The collected data is based on the data collected from inpatients, Therefore, your conclusion is limited to this population.
Round 2
Reviewer 1 Report
The authors answered all my comments except my second comment which was:
The structure of your paper is so weird. Please consider the main structure of the paper at the end of the introduction.
For example, you should mention:
The rest of this paper is organized as follows. The.... is presented in Section 2. Then, the....are proposed in section 3....
Author Response
Thank you for your comment. Our paper has referred to the structure of some latest similar researches in Antibiotics [1-3], including Introduction followed by Materials and Methods, Results, Discussion, and Conclusion sections. This structure is also commonly used by similar studies in other publications [4,5]. To keep it concise, we did not present this part at the end of our Introduction section and hope you could agree.
[1] Meschiari M, Onorato L, Bacca E, Orlando G, Menozzi M, Franceschini E, Bedini A, Cervo A, Santoro A, Sarti M, Venturelli C, Biagioni E, Coloretti I, Busani S, Girardis M, Lòpez-Lozano JM, Mussini C. Long-Term Impact of the COVID-19 Pandemic on In-Hospital Antibiotic Consumption and Antibiotic Resistance: A Time Series Analysis (2015-2021). Antibiotics (Basel). 2022 Jun 20;11(6):826. doi: 10.3390/antibiotics11060826.
[2] Rashid MM, Akhtar Z, Chowdhury S, Islam MA, Parveen S, Ghosh PK, Rahman A, Khan ZH, Islam K, Debnath N, Rahman M, Chowdhury F. Pattern of Antibiotic Use among Hospitalized Patients according to WHO Access, Watch, Reserve (AWaRe) Classification: Findings from a Point Prevalence Survey in Bangladesh. Antibiotics (Basel). 2022 Jun 16;11(6):810. doi: 10.3390/antibiotics11060810.
[3] Friedli O, Gasser M, Cusini A, Fulchini R, Vuichard-Gysin D, Halder Tobler R, Wassilew N, Plüss-Suard C, Kronenberg A. Impact of the COVID-19 Pandemic on Inpatient Antibiotic Consumption in Switzerland. Antibiotics (Basel). 2022 Jun 11;11(6):792. doi: 10.3390/antibiotics11060792.
[4] Klein EY, Milkowska-Shibata M, Tseng KK, Sharland M, Gandra S, Pulcini C, Laxminarayan R. Assessment of WHO antibiotic consumption and access targets in 76 countries, 2000-15: an analysis of pharmaceutical sales data. Lancet Infect Dis. 2021 Jan;21(1):107-115. doi: 10.1016/S1473-3099(20)30332-7.
[5] Ingelbeen B, Phanzu DM, Phoba MF, Budiongo MYN, Berhe NM, Kamba FK, Kalonji L, Mbangi B, Hardy L, Tack B, Im J, Heyerdahl LW, Da Luz RI, Bonten MJM, Lunguya O, Jacobs J, Mbala P, van der Sande MAB. Antibiotic use from formal and informal healthcare providers in the Democratic Republic of Congo: a population-based study in two health zones. Clin Microbiol Infect. 2022 Apr 18:S1198-743X(22)00205-1. doi: 10.1016/j.cmi.2022.04.002.
Reviewer 2 Report
Sometimes the wording consumption can be replaced by "usage". For instance line 52: ...antibiotic "usage" in China.
I propose to simplify the description of references in the manuscript; e.g. line 74-75: healthcare costs...(1-4)...
Author Response
Thank you for your advice. We have revised correspondingly. We hope that it now reads more smoothly and believe that the manuscript is better. If there is anything we still need to improve, please let us know.
Reviewer 3 Report
The quality of the figures is very bad and the manuscript will need to be revised for English
Author Response
Thank you for your comment. We have revised the Figures correspondingly at a sufficiently high resolution and proofed the manuscript by a native speaker accordingly.
Round 3
Reviewer 1 Report
The authors just answered all my comments except the following one:
Please explain the structure of your paper at the end of the introduction. For example:
Part 1 explains the material such as data analysis and collection. Part 2 demonstrates the results....
Please add one paragraph like the abovementioned example at the end of the introduction which provides an outline for the future readers.
Author Response
Thank you for your comment. We have revised correspondingly in the Introduction section.
“The rest of this paper is organized as follows. The methodology of the study was presented in Section 2. Then, the detailed results of FDC antibiotic consumption were proposed in Section 3. In Section 4, we discussed the findings of the study. The conclusion of the study was presented in Section 5.”
Reviewer 3 Report
Thank you for addressing my comments